# *RadA*, a Key Gene of the Circadian Rhythm of *Escherichia coli*

**DOI:** 10.3390/ijms23116136

**Published:** 2022-05-30

**Authors:** Aissatou Bailo Diallo, Soraya Mezouar, Asma Boumaza, Oksana Fiammingo, Benjamin Coiffard, Pierre Pontarotti, Benoit Desnues, Jean-Louis Mege

**Affiliations:** 1Microbes Evolution Phylogénie et Infection, Institut Recherche et Développement, Aix-Marseille University, 13005 Marseille, France; soraya.mezouar@univ-amu.fr (S.M.); asma-fatima.boumaza@outlook.fr (A.B.); oksana.fiammingo@gmail.com (O.F.); bcoiffard.aphm@gmail.com (B.C.); pierre.pontarotti@univ-amu.fr (P.P.); benoit.desnues@univ-amu.fr (B.D.); jean-louis.mege@univ-amu.fr (J.-L.M.); 2IHU-Méditerranée Infection, 13005 Marseille, France; 3CNRS SNC5039, 13005 Marseille, France; 4APHM, UF Immunologie, 13005 Marseille, France

**Keywords:** bacteria, circadian rhythm, *radA*

## Abstract

Circadian rhythms are present in almost all living organisms, and their activity relies on molecular clocks. In prokaryotes, a functional molecular clock has been defined only in cyanobacteria. Here, we investigated the presence of circadian rhythms in non-cyanobacterial prokaryotes. The bioinformatic approach was used to identify a homologue of KaiC (circadian gene in cyanobacteria) in *Escherichia coli*. Then, strains of *E. coli* (wild type and mutants) were grown on blood agar, and sampling was made every 3 h for 24 h at constant conditions. Gene expression was determined by qRT-PCR, and the rhythmicity was analyzed using the Cosinor model. We identified RadA as a KaiC homologue in *E. coli.* Expression of *radA* showed a circadian rhythm persisting at least 3 days, with a peak in the morning. The circadian expression of other *E. coli* genes was also observed. Gene circadian oscillations were lost in *radA* mutants of *E. coli*. This study provides evidence of molecular clock gene expression in *E. coli* with a circadian rhythm. Such a finding paves the way for new perspectives in antibacterial treatment.

## 1. Introduction

Almost all living organisms are subject to natural rhythms controlled by endogenous clocks with a period of nearly 24 h and synchronization by environmental factors [1]. For many years, researchers thought that a biological rhythm could not be present in organisms that multiply faster than the period of that cycle [2]. However, in the 1980s, circadian rhythm and molecular clocks were reported in prokaryote organisms, namely, cyanobacteria ([3,4,5]).

Cyanobacteria are unicellular prokaryotes with a circadian rhythm under the control of KaiA, KaiB, and KaiC protein [6]. The rhythm is generated by a feedback loop with a period of about 24 h. During the day, KaiA interacts with KaiC to induce a signal to active the photosynthesis. During the night, KaiC is activated following the initiation of the nitrogen cycle, inducing the expression of target genes involved in the clock loop (RepA, RepB). Then, KaiB deactivates KaiC at the end of the night and the cycle starts again [2,7]. Thus, the Kai ABC cluster controls the cyclic behaviors of cyanobacteria including cell division, photosynthesis, and adaptation to environmental changes [8].

The presence of circadian rhythms in prokaryotic organisms, notably cyanobacteria, has led researchers to explore the presence of clock genes in non-cyanobacterial prokaryotes. Interestingly, Schmelling et al. found homologues of the cyanobacterial circadian protein KaiC in non-circadian bacteria including *Pseudomonas* sp. and in *Archaea* using a universal BLAST analyses [9]. In addition, it was shown that the circadian clock of *Klebsiella pneumoniae* is driven by temperature [10]. Recently, Eelderink-Chen et al. reported that the non-photosynthetic prokaryote *Bacillus subtilis* presented a circadian clock over a 24 h period [11]. Moreover, in *Escherichia coli*, the heterologous expression of Kai ABC showed a circadian oscillation of KaiC phosphorylation in the presence of KaiA and KaiB, suggesting that non-circadian bacteria have the machinery to generate a circadian rhythm [12].

In this work, we investigated the presence of homologs of the cyanobacterial protein KaiC in non-cyanobacterial prokaryotes by using the bacterium *E. coli* as a study model. Our hypothesis was that circadian oscillations exist in bacteria and control their functions (cell division, metabolism) and their interaction with the host. Just like the eukaryotic host for which the circadian rhythm controls the immune response to infection [13], the bacterial circadian rhythm could allow them to escape the host response and survive environmental changes (e.g., antibiotics).

Thus, by using a bioinformatics method, we identified *radA* gene as a *kaiC* homologue in the bacterium *E. col**i*. Then, using Cosinor analysis, we showed for the first time a circadian rhythm of gene expression in *E. coli*. This study demonstrates the presence of circadian rhythm and molecular clock expression in *E. coli*. 

## 2. Results and Discussion

### 2.1. RadA Is a KaiC Homologue in Escherichia Coli

In order to identify a KaiC homologue in *E. coli*, we used the protein sequence of KaiC identified in *Pseudomonas* sp. and performed a position-specific iterative (PSI) basic local alignment search tool (BLAST) against the *E. coli* proteome by targeting non-redundant protein sequences. We identified RadA as the best match, with 72% coverage, 31.16% identity, and 1.e^−04^ e-value. The Figure 1 shows the alignment of the KaiC and RadA sequences using Clustal Omega [14] (Figure 1). RadA, also known as Sms, is a highly conserved protein in almost all bacteria and plants and is found in the operon *serB-radA-nadR* [15]. Interestingly, RadA belongs to the RecA superfamily of proteins, which also includes KaiC [16,17], and presents a sequence similarity with the recombinase protein RecA [15]. 

### 2.2. RadA and recA Genes Present a Circadian Rhythm in Escherichia Coli

Given the similarity between RadA and KaiC, we then wondered whether the expression of *radA* and *recA* genes followed circadian oscillations. Expression of *radA* and *recA* was measured every 3 h in bacteria cultivated on blood agar over a 24 h period to avoid the effect of bacterial proliferation as illustrated in Figure 2A. Cosinor analysis showed a circadian rhythm for *radA* gene expression over these 24 h (Figure 2B), with significant rhythmic parameters including MESOR (2.33, *p* < 0.001), amplitude (1.21, *p* = 0.01), and acrophase (4.70, *p* = 0.003) (Table 1). To our knowledge, this is the first report that evidences circadian expression of *radA*, suggesting that it may be an *E. coli* homologue of the circadian *kaiC* gene [16] and may be involved in *E coli* circadian oscillations.

Moreover, we showed that *radA* expression peaked during the morning, as previously reported for the *kaiC* gene in cyanobacteria [5]. 

It is important to note that we did not find any correlation between *radA* expression and bacterial biomass during the 24 h period (r^2^ = 0.39; *p* = 0.09) (Figure 2C). 

Interestingly, when bacteria were cultured at constant conditions for 3 days, we found a significant circadian expression of *radA* gene for the 3 consecutive days, suggesting that the rhythm persists over time (Appendix A). Furthermore, the *radA* gene showed a significant circadian rhythm with an amplitude of 0.23 and acrophase occurring at CT20 in *E. coli* bacterium cultured on the LB agar (Appendix A), demonstrating that the rhythm is not specific to blood agar.

We also identified a circadian rhythm for *recA* expression (Figure 2B) with significant rhythmic parameters including MESOR (0.86, *p* < 0.001), amplitude (0.47, *p* = 0.02), and acrophase (8.57, *p* = 0.04) (Table 1). 

We next explored gene expression in *radA* and *recA* mutants, which showed no expression of *radA* and *recA* genes (Figure 2B). No circadian rhythm was observed for *radA* and *recA* genes in ∆*recA* and ∆*radA E. coli* strains, respectively (Figure 2B, Table 1), suggesting a cooperation between RadA and RecA in circadian rhythmicity, which may be reminiscent of the cooperation of RadA and RecA during bacterial homologous recombination [18]. 

Finally, we wondered if the circadian rhythmicity of *radA* and *recA* expression was specific to these genes. We measured the rhythmic expression of genes (*ssrA* and *cysG*) susceptible to be variable and genes (*ihfB* and *rrsA*) that are stable [19]. As expected, we found a significant circadian rhythm for *ssrA* and *cysG* with statistically significant peaks and amplitudes, while the expression of the *ihfB* and *rrsA* genes did not show circadian rhythmicity (Appendix A, Table 1). Interestingly, the cyclic expression of the variable genes *ssrA* and *cysG* was abolished in *radA* mutants, while in *recA* mutants, cyclic expression persisted for *ssrA* and appears for *rrsA* (Appendix A, Table 1). Taken together, our results suggest that the genes encoding RadA and RecA have a cyclic circadian expression and are involved in the regulation of the circadian expression of target genes such as *ssrA* or *cysG*.

To conclude, in this work, we identified *radA* as a probable homologous gene for the *kaiC* clock gene and have shown that its expression as well as that of its partner *recA* and other genes involved in different functions in *E. coli* follow a circadian rhythm. It was long assumed that non-cyanobacterial organisms with a very short life cycle would not need a rhythm [20], but here we showed circadian oscillations in *E coli*. Circadian rhythms in pathogenic bacteria opens several research questions, including their involvement in host response and antibiotic resistance. Recently, Kovac et al. showed that the genetic background of *E. coli* is important for the effect on mammalian circadian clock genes, indicating a possible future use of probiotic strains of *E. coli* to influence the host circadian clock [21]. Therefore, pathogenic bacteria could affect the circadian rhythm of the host immune system through their own rhythm.

However, several questions, including the exact role of *radA* in the generation of this rhythm and the identification of the entire circadian machinery in the bacterium, remain unanswered. In addition, this study has its limitations, including the fact that it does not demonstrate temperature entrainment and compensation. Moreover, although the Cosinor model constituted an appropriate model to evaluate circadian rhythm over a 24 period, further investigations using continuous recordings such as bioluminescence techniques and models able to fit the most accurate period (Fourier transformation or non-linear models) are required to support our study. Further experiments are needed to elucidate those questions, including the function of *radA* and *recA* as clock genes and the identification of other genes involved in the circadian machinery in non-cyanobacterial prokaryotes. This study provides very interesting new information and opens up a wide range of research perspectives and a new therapeutic target for the treatment of bacterial infectious diseases.

## 3. Materials and Methods

### 3.1. Protein Sequence Analysis

Protein sequences for KaiC from *Pseudomonas* sp. 460 were extracted from the Uniprot site (https://www.uniprot.org/ accessed on 5 May 2022). A position-specific iterative (PSI) basic local alignment search tool (BLAST) was used to realize protein sequence against the proteome of *Escherichia coli* targeting non-redundant protein sequences. Percentage identity greater than 30 and e-value less than 0.05 were considered. KaiC and RadA sequences were then aligned using Clustal Omega

### 3.2. Bacterial Culture 

The *Escherichia coli* strain (O157:H7, ATCC) and the *radA* and *recA* mutants in the BW25113 strain (kindly provided by Dr. Laurent Aussel, CNRS, Marseille) were thawed at midnight of day 0, then were spotted on blood (Becton Dickinson 244510, Le Pont de Claix, France) or LB (*Luria-Bertani*) agar on 4 points/agar with 7.10^5^ bacteria per point. Then, bacteria were incubated at 37 °C for 24 h. Colonies were then sampled every 3 h for 24 h (starting at midnight on day 1) at constant temperature in the incubator (without light/dark cycle) and suspended in 4 mL of saline solution (Figure 2A). In some experiments, the bacteria were maintained for 3 days in culture, and colonies were sampled every 3 h. All samples were frozen and stored at −80 °C before RNA extraction. 

### 3.3. Bacterial RNA Extraction and Real-Time Quantitative PCR (q-RTPCR)

Bacterial total RNA was extracted using the QIamp RNA kit (Qiagen, Courtaboeuf, France) according to the manufacturer′s instructions. Briefly, bacterial lysate was precipitated in 70% ethanol, transferred to the columns, and then treated with DNase I (Qiagen) for 15 min at room temperature. The samples were then washed and eluted in RNase-free water. The quantity and quality of RNA were assessed by spectrophotometry (Nano Drop Technologies, Wilmington, DE, USA).

Reverse transcription was performed using the M-MLV transcriptase kit (Life Technologies) and random hexamers (Life technologies, Carlsbad, CA, USA). q-RTPCR was then performed using the Smart SybrGreen kit (Roche Diagnostics, Meylan, France) to determine the expression level of *radA*, *recA*, *ihfB*, *cysG*, *ssrA*, and *rrsA* according to the *rpoB* housekeeping gene [22]. The sequences of the primers are summarized in Table 2. Gene expression was calculated according to the following formula: gene expression = ΔCt or 2^−ΔΔCt^ (fold change); with Ct = cycle threshold, ΔCt = Ct_target_ − Ct*_rpoB_*. For the fold change, we reported the results to the maximum value: ΔΔCt = ΔCt of the time point x–ΔCt of the maximum value. 

### 3.4. Bacterial DNA Extraction and qPCR

DNA was extracted from the bacterial samples using the mini DNA QIamp (Qiagen) kit according to the manufacturer′s instructions. DNA was quantified by spectrophotometry (Nano Drop Technologies) and amplified by quantitative PCR using the Smart SybrGreen kit (Roche Diagnostics) and the single copy *rpoB* gene [22]. Bacterial quantification was determined by using a standard curve generated by successive 10-fold dilution from a mother suspension of known bacterial concentration.

### 3.5. Statistical Analysis 

The analysis of the circadian rhythm was performed using the Cosinor transformation, which allows for the estimation of the variations of a given variable over a 24 h period (estimation with 95% confidence limits of the parameters that characterize a biological rhythm) [13]. A circadian rhythm of gene expression is considered significant if the mesor, amplitude, and acrophase are all statistically significant. The statistical analyses were performed on R studio version 3.1.0. A Pearson′s correlation test was used to analyze the correlation between bacterial biomass and *radA* gene expression. Statistical significance was defined for a *p*-value < 0.05.

## Figures and Tables

**Figure 1 ijms-23-06136-f001:**
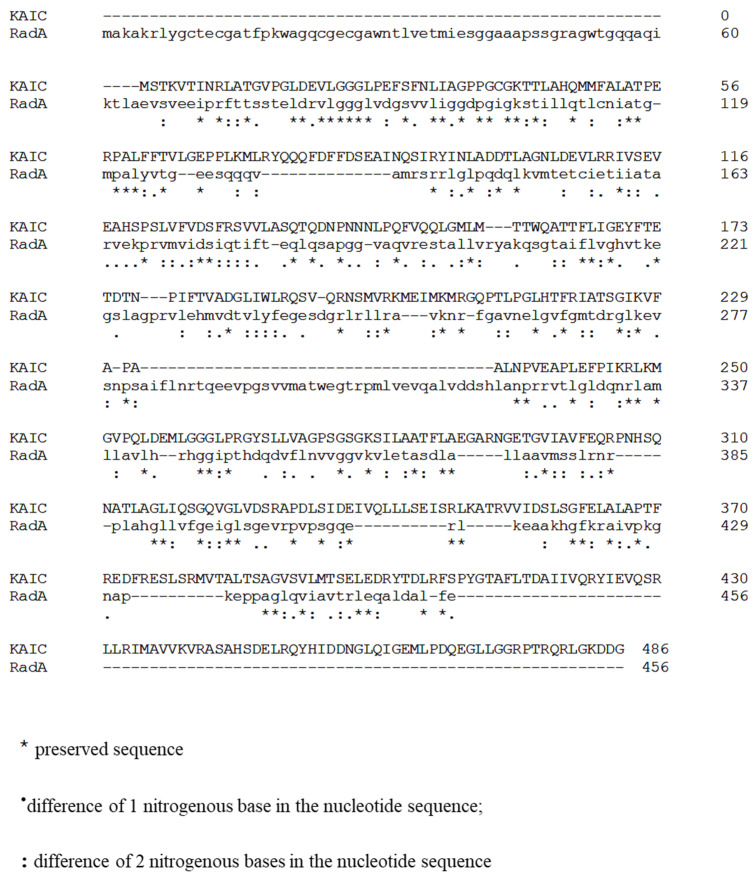
Alignment of KaiC from *Pseudomonas* and RadA sequences. Protein sequence of the circadian clock protein KaiC from *Pseudomonas* sp. 460 were extracted from the Uniprot site, and a position-specific iterative (PSI) basic local alignment search tool (BLAST) was used to realize protein sequence against the proteome of *Escherichia coli* targeting non-redundant protein sequences. The alignment of KaiC and RadA was generated using Clustal Omega (the star represents preserved sequences; one point represents a difference of 1 nitrogenous base in the nucleotide sequence and deux points represents a difference of 2 nitrogenous bases in the nucleotide sequence).

**Figure 2 ijms-23-06136-f002:**
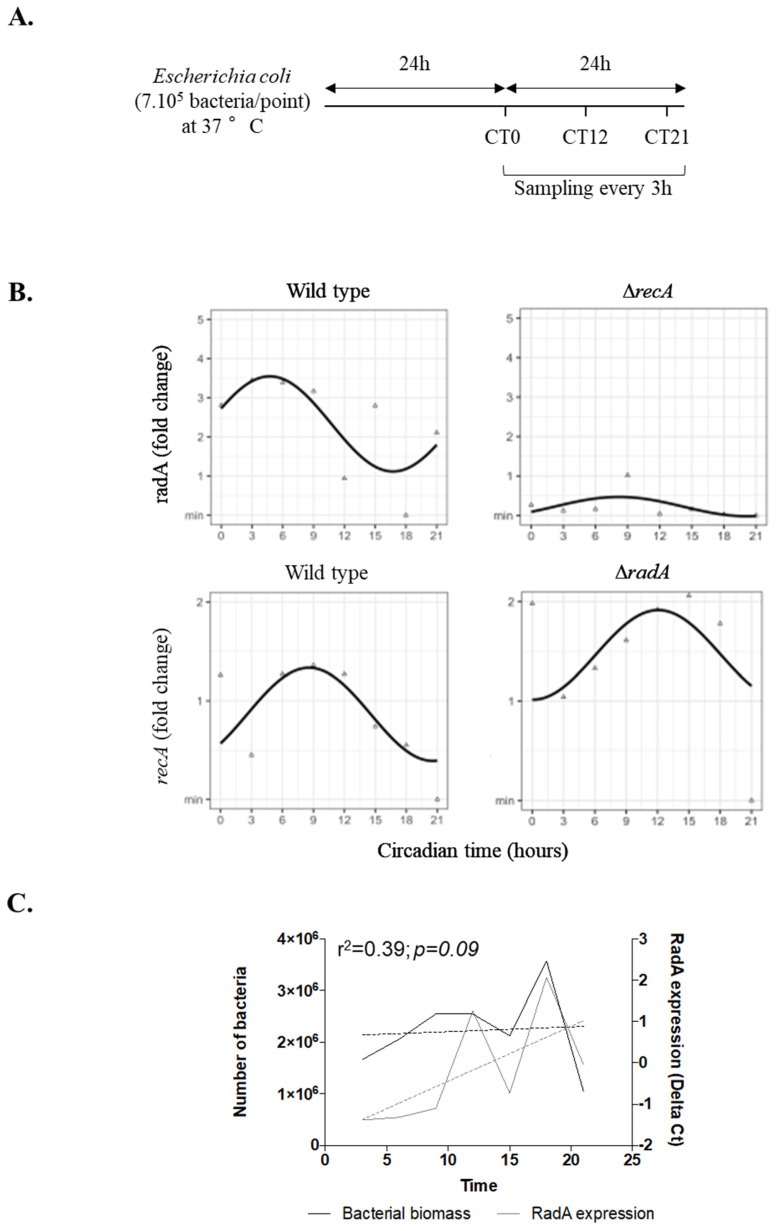
Circadian rhythm of gene expression in *Escherichia coli*. Bacteria (wild-type *E. coli*, *radA* and *recA* mutants) were spotted on blood agar on 4 points/agar with 7. 10^5^ bacteria per point and incubated at 37 °C for 24 h. Colonies were then collected every 3 h for 24 h at a constant temperature (37 °C) without light/dark cycle. (**A**) Schematic representation of the experimental set-up. CT represents the circadian time; the grey bar is daylight, and the black bar is night. (**B**) The expression of the *radA* and *recA* genes was evaluated by q-RTPCR and expressed as fold change using the 2^−ΔΔCt^ formula. The triangles correspond to the raw data without Cosinor transformation; the sinusoids represent the expression of the genes over time after adjustment of the values by the Cosinor model. Statistical analysis was performed using the cosine function in R studio (the results are representatives of three experiments). (**C**) The number of bacteria (bacterial biomass, black) and *radA* gene expression (calculated with ΔCt formula, grey) were evaluated by qPCR and q-RTPCR, respectively. Pearson′s correlation test was used to analyze the correlation between bacterial biomass and *radA* gene expression; the dashed line represents the linear regression. Statistical significance was defined for a *p* value < 0.05.

**Table 1 ijms-23-06136-t001:** Rhythmic parameters (MESOR, amplitude, and acrophase) of investigated genes.

	Gene	MESOR	CI 95%	*p*-Value	Amplitude	CI 95%	*p*-Value	Acrophase	CI 95%	*p*-Value	Significant
**Wild type**	*radA*	2.33	(1.63; 3.02)	**<0.001**	1.21	(0.23; 2.20)	**0.01**	4.70	(1.64; 7.83)	**0.003**	**Yes**
	*recA*	0.86	(0.57; 1.15)	**<0.001**	0.47	(0.06; 0.88)	**0.02**	8.57	(5.25; 11.88)	**0.04**	**Yes**
	*ihfB*	2.10	(1.38; 2.83)	**<0.001**	1.09	(0.07; 2.11)	**0.03**	2.10	(0.00; 5.65)	0.24	No
	*cysG*	1.72	(1.07; 2.37)	**<0.001**	1.49	(0.57; 2.40)	**0.001**	4.20	(1.83; 6.5)	**<0.001**	**Yes**
	*rrsA*	2.05	(1.18; 2.91)	**<0.001**	1.59	(0.37; 2.81)	**0.01**	21.32	(18.4; 24.2)	0.07	No
	*ssrA*	3.98	(2.75; 5.21)	**<0.001**	3.84	(2.10; 5.58)	**<0.001**	16.85	(15.1; 18.6)	**<0.001**	**Yes**
**Δ*radA***	*radA*	-	-	-	-	-	-	-	-	-	
	*recA*	1.36	(1; 1.68)	**<0.001**	0.45	(-0.24; 1.14)	**0.005**	12.1	(6.21; 17.9)	0.98	No
	*ihfB*	1.12	(0.56; 2.92)	**<0.001**	0.42	(-0.37; 1.21)	0.30	19.6	(12.5; 21.2)	0.23	No
	*cysG*	0.32	(0.17; 0.45)	**<0.001**	0.20	(0.00; 0.39)	0.05	19.2	(15.4; 22.9)	**<0.001**	No
	*rrsA*	0.57	(0.17; 0.97)	**0.005**	0.38	(-0.18; 0.94)	0.19	0.91	(19.2; 6.60)	0.75	No
	*ssrA*	1.39	(0.94; 1.84)	**<0.001**	0.83	(0.20; 1.47)	**0.01**	0.72	(21.8; 3.64)	0.62	No
**Δ*recA***	*radA*	0.22	(0.00; 0.45)	**0.05**	0.24	(-0.08; 0.57)	0.14	8.18	(3.17; 13.2)	0.14	No
	*recA*	-	-	-	-	-	-	-	-	-	
	*ihfB*	0.29	(0.09; 0.50)	**0.005**	0.16	(-0.12; 0.45)	0.27	10.3	(3.58; 17.0)	0.62	No
	*cysG*	1.03	(0.58; 1.48)	**<0.001**	0.32	(-0.31; 0.96)	0.32	8.14	(0.67; 15.6)	0.31	No
	*rrsA*	0.96	(0.69; 1.23)	**<0.001**	0.63	(0.25; 1.00)	**0.001**	7.02	(4.70; 9.33)	**<0.001**	**Yes**
	*ssrA*	1.12	(0.80; 1.45)	**<0.001**	0.80	(0.33; 1.27)	**<0.001**	6.66	(4.45; 8.86)	**<0.001**	**Yes**

Data in bold are data with significant *p* value.

**Table 2 ijms-23-06136-t002:** Sequences of primers used in the qPCR.

Gene Symbol	Forward Primer (5′-3′)	Reverse Primer (3′-5′)
** *rpoB* **	GTTTCACCACCATCCCACATTC	TTCGGCGTTACCTTACCAAC
** *radA* **	GTGATGGTGGTATGGGAAGG	GCTAAGTCGGCACTGGTTTC
** *recA* **	ATTCTACGCCTCTGTTCGTCTC	GCATTCGCTTTACCCTGACC
** *rrsA* **	CTTACGACCAGGGCTACACA	CTTGTTACGACTTCACCCCAGT
** *ssrA* **	GTTACACATTGGGGCTGATTCT	CTTTTGGGTTTGACCTCTCTTG
** *ihfB* **	GGTTTCGGCAGTTTCTCTTTG	CCAGTTCTACTTTATCGCCAGTC
** *cysG* **	AGCGTTTATTCCACAGTTCACC	CGTTACAGAAGATGCGACGAG

## Data Availability

Not applicable.

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
