# Peer review of "RadA, a Key Gene of the Circadian Rhythm of Escherichia coli"

_ijms, 2022, doi:10.3390/ijms23116136_

Round 1

Reviewer 1 Report

In the present study, Diallo et al. investigated the presence of circadian rhythms in non-cyanobacteria species. In the results section, the authors showed the homologue RadA as KaiC in E. coli. It was reported that E. coli has a gene for circadian oscillations. Lastly, this study provide support for the idea that molecular clock gene expression in E. coli with a circadian rhythm. The biological question is really good to work on and is worth finding the answers to. However, the study has major limitations. Please find the comments below.

Authors have provided evidence regarding the circadian rhythms in non-cyanobacteria species. In this case, the model organism is E. coli, but they lack the major point to introduce circadian rhythms in prokaryotes. The introduction is limited and provides sufficient background on the topic. I would strongly recommend that authors provide the details and meaning of circadian rhythms in prokaryotes. Define what circadian rhythms are in non-cyanobacteria species.

The identity determined between the RadA gene and kaiC is 31.16%. However, E. coli contains 4448 protein-coding genes. Why do the authors want to show only the RadA gene as the circadian rhythm specific gene? How the authors want to justify that there are not any more genes which could be more relevant than RadA for circadian rhythms.

It is important that authors report which cluster/operon RadA was involved in E. coli. This will help to determine the more relevant gene responsible for circadian rhythms in E. coli.

How is it possible to extrapolate or claim the information of one gene expression to correlate with circadian oscillations? Could authors please justify this concept? The authors themselves reported that the Kai cluster is involved in cyanobacteria.

Author Response

In the present study, Diallo et al. investigated the presence of circadian rhythms in non-cyanobacteria species. In the results section, the authors showed the homologue RadA as KaiC in E. coli. It was reported that E. coli has a gene for circadian oscillations. Lastly, this study provide support for the idea that molecular clock gene expression in E. coli with a circadian rhythm. The biological question is really good to work on and is worth finding the answers to. However, the study has major limitations. Please find the comments below.

 Authors have provided evidence regarding the circadian rhythms in non-cyanobacteria species. In this case, the model organism is E. coli, but they lack the major point to introduce circadian rhythms in prokaryotes. The introduction is limited and provides sufficient background on the topic. I would strongly recommend that authors provide the details and meaning of circadian rhythms in prokaryotes. Define what circadian rhythms are in non-cyanobacteria species.

We agree with the reviewer. The introduction has been modified and the details of the circadian rhythms of prokaryotes and our hypothesis about the definition of rhythms in non-cyanobacterial organisms have been added (see the section introduction)

 The identity determined between the RadA gene and kaiC is 31.16%. However, E. coli contains 4448 protein-coding genes. Why do the authors want to show only the RadA gene as the circadian rhythm specific gene? How the authors want to justify that there are not any more genes which could be more relevant than RadA for circadian rhythms.

We agree with the reviewers on the percentage identity between RadA and KaiC. Indeed, we do not claim that only RadA is specific for the E coli circadian rhythm, . Moreover, in this study, we were also interested in RadA's partner (RecA) (see figure 2) and other E coli genes (see figure S2 and results section). For all the organisms in which circadian oscillations have been demonstrated, it has been shown that they are generated following the expression of several genes, most often in a feedback loop (Cohen et al 2015, Diallo et al, 2020). We therefore believe that non-cyanobacterial prokaryotes also have a complete circadian machinery with several genes involved. Only in this communication, we are interested in the KaiC homolog. Further experiments are needed to identify the full set of circadian rhythm genes in non-cyanobacterial prokaryotes and their mechanism.

 It is important that authors report which cluster/operon RadA was involved in E. coli. This will help to determine the more relevant gene responsible for circadian rhythms in E. coli.

We agree with the reviewers. RadA is involved in the operon serB-radA-nadR. This was added in the text (see lines 82-83)

 How is it possible to extrapolate or claim the information of one gene expression to correlate with circadian oscillations? Could authors please justify this concept? The authors themselves reported that the Kai cluster is involved in cyanobacteria.

We thank the reviewers for this question. Indeed, in this study we showed not only that radA is a homolog of KaiC in E coli, but also that it had a circadian expression. Other genes in E coli (recA, ssrA and cysG) have shown a statistically significant circadian oscillation over a 24h period. We thus show the presence of circadian rhythm in E coli and that RadA could be one of the clock genes of this rhythmicity but not the only one. Further experiments are needed to identify the full set of circadian rhythm genes in E coli.

Reviewer 2 Report

Diallo et al discuss the circadian rhythm of E.coli. The manuscript is very brief and would benefit with a more in depth discussion of the results and impact.

  1. The introduction is very brief and could benefit from some expansion about the importance/relevance of this work and explanation of the methods used.
  2. Line 65: some French has slipped into the text. Please double check that the manuscript is written in grammatically correct English.
  3. Line 66 - I think you meant to indicate,  - please fix
  4. The formatting of Table 2, table s1 and s2 could be improved - I suggest making it "landscape" for clarity. The significant column is redundant since you provide the p values.
  5. The discussion does not adequately discuss the impact and relevance of this work in the broader field. Expansion is required.
  6. Section 3.2 - bacterial culture and maintenance details should be provided in detail, along with preparation details for the blood agar.
  7.  

Author Response

Diallo et al discuss the circadian rhythm of E.coli. The manuscript is very brief and would benefit with a more in depth discussion of the results and impact.

 The introduction is very brief and could benefit from some expansion about the importance/relevance of this work and explanation of the methods used.

We thank the reviewers for this comment. The introduction has been enlarged and the importance of this work and the methods used has been added.

  1. Line 65: some French has slipped into the text. Please double check that the manuscript is written in grammatically correct English.

We thank the reviewers for this comment. Those errors were corrected.

  1. Line 66 - I think you meant to indicate- please fix

We thank the reviewers for this comment. Those errors were corrected.

  1. The formatting of Table 2, table s1 and s2 could be improved - I suggest making it "landscape" for clarity. The significant column is redundant since you provide the p values.

We thank the reviewers for this remark. The table’s format has been corrected and the significant column suppressed.

  1. The discussion does not adequately discuss the impact and relevance of this work in the broader field. Expansion is required.

We thank the reviewer for this remark. The discussion has been expensed (see section results and discussion)

  1. Section 3.2 - bacterial culture and maintenance details should be provided in detail, along with preparation details for the blood agar.

The details of bacteria culture were added (see lines 148 – 154). Blood agar plates were bought ready-to-use, the reference of the provider has been added to the text (see line 155).

Round 2

Reviewer 1 Report

No more comments.

This manuscript is a resubmission of an earlier submission. The following is a list of the peer review reports and author responses from that submission.